# Potential Impact of Human Cytomegalovirus Infection on Immunity to Ovarian Tumours and Cancer Progression

**DOI:** 10.3390/biomedicines9040351

**Published:** 2021-03-30

**Authors:** Momodou Cox, Apriliana E. R. Kartikasari, Paul R. Gorry, Katie L. Flanagan, Magdalena Plebanski

**Affiliations:** 1School of Health & Biomedical Sciences, RMIT University, Melbourne 3082, Australia; momodou.cox@student.rmit.edu.au (M.C.); april.kartikasari@rmit.edu.au (A.E.R.K.); paul.gorry@rmit.edu.au (P.R.G.); katie.flanagan@ths.tas.gov.au (K.L.F.); 2School of Medicine and School of Health Sciences, University of Tasmania, Launceston 7250, Australia; 3Department of Immunology and Pathology, Monash University, Melbourne 3004, Australia

**Keywords:** human cytomegalovirus, ovarian cancer, cancer progression, inflammation, immunosuppression

## Abstract

Ovarian cancer (OC) is one of the most common, and life-threatening gynaecological cancer affecting females. Almost 75% of all OC cases are diagnosed at late stages, where the 5-year survival rate is less than 30%. The aetiology of the disease is still unclear, and there are currently no screening method nor effective treatment strategies for the advanced disease. A growing body of evidence shows that human cytomegalovirus (HCMV) infecting more than 50% of the world population, may play a role in inducing carcinogenesis through its immunomodulatory activities. In healthy subjects, the primary HCMV infection is essentially asymptomatic. The virus then establishes a life-long chronic latency primarily in the hematopoietic progenitor cells in the bone marrow, with periodic reactivation from latency that is often characterized by high levels of circulating pro-inflammatory cytokines. Currently, infection-induced chronic inflammation is considered as an essential process for OC progression and metastasis. In line with this observation, few recent studies have identified high expressions of HCMV proteins on OC tissue biopsies that were associated with poor survival outcomes. Active HCMV infection in the OC tumour microenvironment may thus directly contribute to OC progression. In this review, we highlight the potential impact of HCMV infection-induced immunomodulatory effects on host immune responses to OC that may promote OC progression.

## 1. Background

Ovarian cancer (OC) is one of the most prevalent, aggressive, and life-threatening gynaecological malignancies affecting females. Ovarian tumours arise from either epithelial cells, stromal cells, or germ cells. Amongst them, epithelial OC constitutes more than 90% of malignant OC in developed countries [1,2]. About 75% of all OC cases are diagnosed at an advanced stage (stage 3–4) [3], where the 5-year survival rate is less than 30% [2]. The current standard of care for women diagnosed with OC is primary debulking surgery followed by platinum-based chemotherapy with paclitaxel and carboplatin. However, almost 66% of patients experience recurrence within two years of diagnosis and the majority of these recurrences are accounted for by patients diagnosed at late stages [4].

Presently, the early features of ovarian carcinogenesis remain unclear, and there is no effective early detection test or screening method to date [5]. Therefore, the identification of women with the disease is based on their clinical presentation, in combination with currently approved, albeit non-specific, tests including ultrasound and/or serum cancer antigen 125 (CA-125) and human epididymis 4 (HE4) levels, followed by histological confirmation with tissue biopsy [6]. Age, family history, as well as genetic mutations in *breast cancer* (*BRCA)1* or *BRCA2* genes are known risk factors for OC. *BRCA1* and *BRCA2* produce tumor suppressor proteins that help repair damaged DNA and their mutations are inherited in an autosomal dominant pattern [7,8,9]. Inheritance of such harmful mutations causes a compromised ability for cells to repair DNA damage, which leads to additional genetic alterations and ultimately cancer [10,11]. Additionally, women over the age of 45 years have a higher risk of developing OC partly due to an accumulation of somatic mutations during aging [12].

A combination of the classical OC risk factors highlighted above might create a tumour-favouring cellular environment, where some oncogenic viruses may reside and enhance their oncogenic capability. Even though the classification of human cytomegalovirus (HCMV) as an oncogenic virus remains controversial, there is strong emerging evidence showing a prevalence of HCMV infection in breast, colon, prostate, liver, cervical, and brain cancer patients [13]. In a study by Taher et al. [14], HCMV proteins and/or nucleic acids were detected in 98% of breast cancer derived metastatic brain tumours but not in healthy tissues surrounding HCMV-infected brain tumours, suggesting a potential association between HCMV infection and metastatic cancer. Despite the fact that infection with HCMV is rarely associated with OC, recent studies have shown that OC patients with high expression of HCMV-immediate early (IE) and HCMV-pp65 proteins by their ovarian tumours have shorter survival outcomes compared with those with little tumour expression of these proteins [15,16]. These data suggest that infection with HCMV could potentially promote cancer progression. Once HCMV infects host cells, it begins to counteract key anti-viral immune mechanisms needed to control the infection through secretion of immunosuppressive cytokines and impairment of cell-mediated immune responses that are also important in controlling tumour growth. Hence, the focus of this review is to highlight the potential impact of HCMV infection on immune responses to OC that may also promote cancer progression.

## 2. Human Cytomegalovirus (HCMV)

Human cytomegalovirus (HCMV), also known as human herpes virus 5 (HHV-5), infects about 83% of the world’s population, approaching 100% in developing countries [17]. Following primary infection as shown in Figure 1, HCMV establishes a life-long chronic latency in humans [17], primarily in the cluster of differentiation (CD)34^+^ hematopoietic progenitor cell population located in the bone marrow [18]. Latent infection is generally asymptomatic in immunocompetent individuals although symptomatic reactivation can occur, particularly in the immunocompromised or cancer patients [17]. High level circulating pro-inflammatory cytokines are the hallmark of latent HCMV reactivation, particularly when the CD34+ progenitor cells differentiate into inflammatory monocytes or infiltrating macrophages or dendritic cells, which then spread the virus to peripheral organs and body tissues thereby infecting and replicating in a broad number of cell types [18].

## 3. Effect of HCMV Infection on Innate and Adaptive Immune Response

Upon successful entry into host cells, HCMV begins to counteract various host immune response mechanisms needed to control the infection [19] as shown in Figure 2 below. Firstly, the tegument protein pp65 of HCMV represents the major component of mature virus particles that interferes with interferon regulatory factor 3 (IRF3) signalling that activates interferon-induced genes, by reducing IRF3 phosphorylation status and inhibiting its nuclear localisation. HCMV pp65 also downregulates nuclear factor kappa B (NF-κB) activation, further contributing to reduced type 1 interferon (IFN) production, the primary anti-viral cytokines [20]. Secondly, the recognition of HCMV’s early viral proteins (such as intermediate early (IE) proteins 1 and 2) by CD8^+^ T cells is nullified because the HCMV pp65 gene aids in the sequestration of HCMV- immediate early (IE)-1 proteins making them inaccessible to the CD8^+^ T cell pool [20]. Thirdly, HCMV has unique short (US) gene regions within its genome encoding specific gene products (US2, US3, US6, US10, and US11) that contribute to HCMV-mediated the major histocompatibility complex (MHC)-I downregulation, thereby leading to the decreased presentation of HCMV-specific peptides to CD8^+^ T cells [21,22]. Fourthly, to evade cytotoxicity-mediated by natural killer (NK) cells, HCMV expresses unique long (UL) proteins, such as pUL18 (MHC-I viral homolog) and pUL40, that help to inhibit NK cell cytotoxic responses via a “missing self” mechanism [20]. The viral protein pUL18 binds to β2-macroglobulin and then engages with leukocyte immunoglobulin-like receptor 1 (LIR-1), a class I MHC receptor related to killer inhibitory receptors (KIRs), to induce inhibitory signals preventing NK cell-mediated cytolysis. Additionally, pUL40 binds to MHC class I antigen/human leukocyte antigen (HLA)-E and stabilize its surface expression thus amplifying its interaction with NK cell inhibitory receptor, CD94/natural killer group (NKG)-2A (CD94/NKG2A). Fifthly, it is generally acknowledged that interleukin (IL)-10 produced by regulatory T cells (Tregs) plays an important role in suppressing immune responses during HCMV infection. HCMV also produces cmv-IL10 or pUL10, a homolog of human IL-10 that has immunosuppressive properties similar to human IL-10 [23]. Lastly, a study by Wagner et al. [24] has shown that HCMV-derived protein UL18 stimulates the development of an immature phenotype of dendritic cells (DC) by interfering with CD40 ligand-induced maturation of DCs, resulting in reduced MHC-class II expression on immature monocyte-derived DCs. This could potentially reduce the activation of CD4^+^ T-cell responses as well as decrease the elimination of virus-infected cells or tumours by CD8^+^ cytotoxic T cells.

Chronic infection with HCMV causes huge clonal expansion of the CD8^+^ T cell compartment, but a lesser expansion of CD4^+^ T cells, leading to inversion of the normal CD4:CD8 ratio to less than 1 [17]. The HCMV-driven expanded CD8^+^ T cell subset is usually terminally differentiated due to their high expression of CD57 (a marker of differentiation) and loss of expression of CD27 (a marker of activation) and CD28 (co-stimulatory molecule). The combination of chronic HCMV infection, terminally differentiated T cells and inverted CD4:CD8 T cell ratio results in an “immune risk profile” (IRP) that has been strongly associated with immunosenescence and early death in the elderly [25]. Moreover, Bennett et al. [26] also proposed that HCMV-driven inflammation is associated with aging, so called “inflammaging”. Besides changes in CD4^+^ and CD8^+^ T cells, infection with HCMV also influences NK cell differentiation, activation, and receptor expression. Goodier et al. [27] have shown that HCMV infection is associated with rapid phenotypic and functional differentiation of NK cells (CD57+ NK cells), the majority of which express the activating CD94/NKG2C receptor. Expansion of NKG2C+ NK cells can also be achieved by co-culture with HCMV-infected fibroblasts [28].

## 4. Anti-Tumor Immunity in Ovarian Cancer

The human immune system employs various host-protective and anti-tumour mechanisms to prevent ovarian tumours from developing (Figure 3A). Of these, cell-mediated cytotoxicity is the most effective mechanism employed by the immune system to help prevent the establishment of OC and it involves two main cell types: CD8^+^ cytotoxic T cells (CTLs) and natural killer (NK) cells [29,30]. CTLs perform their effector mechanisms in two steps: MHC class I interaction with T cell receptor (TCR) followed by granule mediated killing. Upon MHC-I antigen recognition by TCR on CTLs, a polarisation of the CTL occurs that ensures a level of organization between the CTL and target cell. The CTL undergoes morphological changes and expresses lytic granules such as perforin and granzymes (granule enzymes), which are released to kill target cells [31,32]. Perforin polymerizes to form pores in the target cells, which allow granzymes such as granzyme B to gain entry into target cells [33]. Granzyme B has complex effects within the target cell and promotes apoptosis via BID (a BH3 domain-containing proapoptotic Bcl2 family member) and activation of caspase, a family of protease enzymes involved in programmed cell death [34]. Data from previous studies have shown that the immune system of OC patients is greatly impacted by the developing tumour (Figure 3B) due to the high presence of Tregs in the tumour microenvironment (TME) that classically inhibit CTL responses [35]. Recent studies have shown that the TME’s immune status, including the presence of pro-inflammatory cytokines and Tregs, and the absence of tumour-infiltrating CD8^+^ T cells are all strongly correlated with OC recurrence [35,36,37]. Conversely, the presence of tumour-infiltrating CD8^+^ T cells and a high CD8^+^ T cell/Treg ratio is associated with substantially better survival outcomes, thus highlighting the importance of CTL-mediated immune responses in OC [38].

Conventionally, NK cells, comprising of about 5–10% of peripheral blood lymphocytes, are considered to be part of the innate immune system and have the capacity to kill tumor cells without the need for prior sensitization [39]. Previous studies have shown that NK-mediated effector functions are highly regulated by a balance between inhibitory and activation signals. NK cells express inhibitory receptors such as KIRs that recognize MHC class I on target cells and deliver inhibitory signals to suppress NK cells function [40,41,42]. As tumour cells express little or no MHC class I molecules, they become highly susceptible to NK-mediated cytolysis in what is referred to as the “missing self” hypothesis. Compared with CD8^+^ T cells, a study by Webb et al. [43] showed that infiltrating NK cells expressing the tissue-resident memory marker CD103 (CD103^+^ NK cells) are most often found with CD8^+^ T cells and these cells were the best predictor of positive survival outcomes in primary OC. Besides CD8^+^ T cells and NK cells, the presence of tumour-infiltrating mature and activated dendritic cells (DCs) have been shown to correlate with a better prognosis in OC. They express CD107a, a marker of activation, and help attract more anti-tumour cell such as CTLs and NK cells to the TME [44].

## 5. Immune Homeostasis

An overly reactive immune system is also detrimental as it could result in tissue damage if it fails to resolve. To minimize such damage, the immune system utilizes immune checkpoint inhibitory pathways that are essential for ensuring self-tolerance and moderating the extent and magnitude of effector responses of CTLs and NK cells. These inhibitory pathways involve surface inhibitory receptors such as cytotoxic T lymphocyte antigen-4 (CTLA-4) and programmed cell death protein 1 (PD-1; CD279). They are usually transiently expressed on activated T cells, B cells, macrophages, dendritic cells and Tregs under normal conditions but their increased and prolonged expression is a sign of T cell exhaustion [45]. PD-1 is more extensively expressed on activated cells than CTLA-4 and predominantly modulates effector CTL responses upon interaction with its ligand, programmed death-ligand 1 (PD-L1; B7-H1; CD274) and/or PD-L2 (B7-DC; CD273) on cancer cells (Figure 3B). Subsequently, their interaction generates potent inhibitory signals that inhibits kinases involved in T cell activation [46,47,48,49,50]. The CTLA-4 co-inhibitory receptor is constitutively expressed on T cells and competes with CD28 co-stimulatory receptor for binding to their cognate ligands CD80 (B7.1) and CD86 (B7.2) on cancer cells, for which it has a greater affinity, thereby effectively inhibiting CTL activation [51,52]. Other immune checkpoint inhibitory receptors such as lymphocyte activation gene-3 (LAG-3), T-cell immunoglobulin-3 (TIM-3), band T lymphocyte attenuator (BTLA) and T-cell immunoglobulin and ITIM domain (TIGIT) are also expressed on exhausted T cells and help regulate effector responses of CTLs [53].

## 6. HCMV Infection in Ovarian Cancer

Although HCMV infection is rarely associated with OC, a study by Shanmughapriya et al. [54] detected HCMV-gB by polymerase chain reaction (PCR) in approximately 50% of OC tissues, of which 80% were late stage invasive tumours, suggesting that HCMV infection in the TME may promote cancer progression or metastasis. More recently, another study assessed the presence of HCMV within OC tissue specimens obtained from diagnostic excisional biopsy pre-chemotherapy (DEBPC) and interval debulking surgery (IDS) after neoadjuvant chemotherapy (4–5 times Taxol/Paraplatin). In this study, OC patients with high levels of HCMV-IE and tegument pp65 proteins in their tumours had lower median overall survival compared to those with lower levels or no detectable HCMV proteins [15]. A second recent study by Radestad et al. [16] also investigated the prevalence of HCMV in ovarian cancer and its relation to clinical outcome. In their study, HCMV-IgG levels, HCMV-IE proteins, and pp65 proteins were all shown to be higher in OC patients with malignant or benign cystadenoma (benign ovarian epithelial tumour) compared with age-matched controls. Additionally, OC patients with focal HCMV-pp65 expression in their tumours accompanied with high IgG levels against HCMV were found to live longer when compared with patients showing high expression of HCMV-pp65 protein in their tumours [16]. These findings suggest a possible impact of HCMV infection on immune responses to ovarian tumours. It is important to highlight that the existence of an active HCMV infection with protein production is a rare occurrence in tissues of healthy individuals. Therefore, the presence of an active HCMV infection with HCMV-pp65 and HCMV-IE proteins on ovarian tumours is quite intriguing and needs further investigation in a larger cohorts of OC patients. This would help justify whether administering anti-HCMV treatment to OC patients experiencing active HCMV reactivation in their TME is needed in future personalised treatment approaches for such patients.

## 7. Significance of HCMV Infection in the OC TME

The OC tumour microenvironment (TME) harbors high numbers of pro-tumour cells such as myeloid derived suppressor cells (MDSCs), tumour-associated macrophages (TAMs) and Tregs compared to anti-tumour cells such as CTLs and NK cells. MDSCs and TAMs (M2 type) classically inhibit activation and cytotoxic functions of CTLs and NK cells through production of anti-inflammatory cytokines such as transforming growth factor-β (TGF-β) and interleukin (IL)-10 (Figure 3A). Indeed, TGF-β induces Treg development and activation, thus promoting an increased Treg presence in the OC TME [35,55]. As mentioned earlier, the presence of a higher ratio of Tregs to CTLs in the TME is known to be a poor prognostic indicator [56] and could be influenced by active HCMV infection in the TME.

HCMV infection is known to promote a cellular secretome that modulates its micro-environment by increasing the production of relevant immunosuppressive mediators, such as TGF-β-expressing Tregs and IL-10 [18]. These immunosuppressive mediators (Figure 4) may contribute to OC progression by inhibiting NK cell and CTL effector functions. HCMV viral protein cmvIL-10, a human IL-10 homologue further boosts this immunosuppressive activity by stimulating the maturation of immunosuppressive macrophages [57], and inhibiting DC maturation [58]. In breast cancer, cmvIL-10 was shown to directly promote growth and migration of breast cancer cells [59] and upregulate proto-oncogene Bcl-3 [57]. Together, these data suggest that active infection by HCMV promotes secretion of suppressive immunomodulatory mediators and the release of cmv-IL10 that may modulate T cell immune function and OC progression.

Since Tregs also recognise the same antigens as CD8^+^ T cells, their increased presence in the TME could result in diminished immune responses to ovarian tumour antigens, and thus contribute to tumorigenic tolerance and immune evasion that may further promote OC progression. In context with current literature, Tregs consist of different subsets of immunosuppressive cells arising from CD4^+^ and CD8^+^ T cells [60]. While CD4^+^ Tregs have been extensively studied, the lack of universal markers to distinguish CD8^+^ Treg cells from conventional CD8^+^ T cells means that the function of CD8^+^ Tregs in cancer remains undefined [61]. Zhang et al. [62] identified the expression of Treg markers in CD8^+^ T cells isolated from peripheral blood and fresh tumour tissues of OC patients. Their study found a higher percentage of CD8^+^ Tregs (defined as CD8^+^ CD25^+^ FOXP3^+^ CD28^−^ CTLA-4^+^) in OC patients compared with benign ovarian tumour patients and healthy controls. Furthermore, in the same study, high TGF-β1 levels also correlated positively with the percentage of CD8^+^ Tregs and triggered the suppressive function of in vitro-induced CD8^+^ Treg cells. The expression levels of FOXP3 in CD8^+^ T cells were found to be positively associated with the stage of tumour in OC patients which implies that the OC TME may have the capacity to convert CD8^+^ effector T cells into CD8^+^FOXP3^+^ suppressor cells in vivo [62]. These findings suggest that CD8^+^FOXP3^+^ Tregs may also contribute to antitumor immunity and OC progression, further highlighting their valuable role as potential predictors of clinical outcomes in OC patients. As stated earlier, HCMV-infected cells also secrete TGF-β to suppress effector T cell responses. Thus, HCMV-infected cells in TME might also augment the conversion of CD8^+^ effector T cells into CD8^+^FOXP3^+^ suppressor cells, thereby shifting the immune balance within the TME from cytotoxic to an immunosuppressive state. Thus, a strategy to reverse the immunosuppressive HCMV-infected T cells could help enhance anti-tumour responses against HCMV-infected OC cells.

As mentioned earlier (Figure 2), some of the immune evasion mechanisms employed by HCMV to modulate host immune responses and establish latency can also exacerbate OC progression (Figure 5). A classic example is HCMV UL18 (MHC class I homologue) that helps to counteract NK cell cytotoxicity by binding to the NK cell inhibitory receptor NKG2A/CD94 [63]. In addition, the expression of HLA-E (a non-classical MHC class I protein) is upregulated on HCMV infected cells or tumours by HCMV UL40 that could further inhibit NK cell cytotoxicity against HCMV-infected ovarian tumours [20]. One of the key mechanisms by which NK cells perform their cytotoxic functions involves the interaction of TNF-related apoptosis-inducing ligand (TRAIL) with their cognate death receptors, TRAIL-R1 and TRAIL-R2. Clustering of the TRAIL-R1 and TRAIL-R2 through interaction with TRAIL results in their oligomerization, inducing development of the death-inducing signalling complex, and ultimately, caspase activation and cellular apoptosis [64]. However, HCMV glycoprotein UL141 has been shown to bind TRAIL death receptors directly [64] and thereby inhibiting the TRAIL-mediated killing of HCMV-infected ovarian tumours. Moreover, HCMV UL141 also downregulates the cell-surface expression of CD155, the ligand for the NK-cell-activating receptors DNAM1 and also targets the alternative DNAM1-activating ligand, CD112 [65].

## 8. Association between Human Leucocyte Antigens, γ Markers and Killer Immuno-Globulin-Like Receptors with Human Cytomegalovirus Infection

As described previously, NK cells help provide a major defence against primary HCMV infection through the interaction of their activating and inhibitory KIRs, and HLA class I molecules. Additionally, antibodies (humoral immunity) also play a key role in initiating NK cell responses to HCMV via antibody-dependent cell-mediated cytotoxicity (ADCC) that is known to be influenced by γ marker (GM) immunoglobulin allotypes [66]. GM allotypes are allelic hereditary variants encoded by autosomal codominant alleles that follows a Mendelian law of heredity, which are expressed on immunoglobulin constant region of γ1, γ2 and γ3 chains [66]. Several mechanisms have been suggested to account for the role GM allotypes play in the control of HCMV infections, which includes their ability to modulate the strength of ADCC and the avidity of the FcγR–IgG interaction thereby influencing the effectiveness of the immune responses [67]. A study by Pandey et al. [68] investigated the contribution of GM3/17 allotypes to the magnitude of antibody responses to HCMV glycoprotein B (gB), a component of the viral envelope that is required for HCMV infectivity to host cells. Their results showed GM3/17 allotypes at the γ1 locus, determined the level of HCMV-specific IgG antibodies to gB. However, the exact role of HCMV-specific antibodies in the control of HCMV infection in vivo remains unclear.

In order to provide further insight into the immune mechanisms controlling HCMV, a recent study investigated the interaction between HLA/KIR genes and GM allotypes in the control of HCMV infection within a Sicilian population. Di Bona et al. [69], assessed whether specific KIRs and HLA repertoire could influence the risk of developing symptomatic or asymptomatic disease upon primary HCMV infection. In their study, sixty immunocompetent patients with primary symptomatic HCMV infection were genotyped for KIR and their HLA ligands, along with sixty other individuals with a previous asymptomatic infection as controls. Their results showed that symptomatic patients had a significantly higher frequency of the homozygous A haplotype, which produces more of the inhibitory KIRs thus inhibitting NK cell function. Additionally, they also showed that symptomatic HCMV disease was associated with the HLABw4^T^ allele, whereby the gene product is the ligand of the inhibitory KIR. In a more recent study [67], the same immunocompetent patients with primary symptomatic HCMV infection were genotyped for GM3/17 and GM23 allotypes, along with the participants with a previous asymptomatic infection used as controls. The results of this study further revealed that the individuals carrying the GM23 allotypes (both homozygous and heterozygous), GM17/17, HLA-C2 and Bw4^T^ KIR-ligand groups were associated with the risk of developing symptomatic HCMV infection. Based on these findings, it is reasonable to speculate that OC patients that carry the same genotypic traits as mentioned above are at a higher risk of experiencing HCMV-induced OC progression. It further highlights the valuable role of HLA, KIRs and GM genotyping not only as potential predictors of clinical outcomes of HCMV-infected OC patients, but also in guiding our efforts to generate highly effective personalised immunotherapeutic strategies to improve OC survival outcomes.

## 9. Potential Modulation of Intrinsic Inhibitory Receptors by HCMV

The PD-1/PD-L1 pathway represents a mechanism employed by OC cells to evade endogenous anti-tumour adaptive immune responses. PD-L1 is commonly overexpressed on tumour cells in the TME and its ligation with PD-1 receptors on activated T cells leads to the inhibition of effector CTL responses [51], thus contributing to immune escape mechanisms associated with poor survival outcomes [53,70,71]. Therapeutic blockade of the PD-1/PD-L1 pathway has been shown to mediate tumour eradication with impressive clinical results [72]. A classic example in OC is durvalumab (anti-PD-L1), a human immunoglobulin G1 kappa monoclonal antibody that works by blocking the interaction of PD-L1 with PD-1 and CD80 thereby promoting antitumour immune responses [73]. Since it is an engineered monoclonal antibody (mAb), it does not induce antibody-dependent cellular cytotoxicity or complement-mediated cytotoxicity. To date, durvalumab has been given to many OC patients as part of ongoing studies, either as monotherapy or in combination with other anticancer agents. Combination therapy of durvalumab with olaparib (a drug that prevents DNA repair) induced objective responses in more than 70% of patients with relapsed, platinum-sensitive, *BRCA*-mutated OC [74].

In recent years, many studies have shown that PD-1 expression on the surface of CD4^+^ and CD8^+^ T cells is increased upon HCMV infection or reactivation [75,76,77,78]. A more recent study by Pesce et al. [79] identified a subset of fully mature NK cells expressing high PD-1 (PD-1^bright^) in peripheral blood of HCMV-seropositive healthy individuals. The proportions of these PD-1^bright^ NK cells were also shown to be higher in the ascites of a cohort of OC patients, suggesting their possible expansion in TMEs. Functional analysis revealed a reduced proliferative capability and impaired antitumour activity that was partially restored by antibody (anti-PD-L1/2 mAb) mediated disruption of the PD-1/PD-L interaction [79]. Recently, many studies have reported elevated expression levels of PD-L1 on glioma cells in glioblastoma (GBM) [80]. GBM is the most common adult primary brain tumour characterised by highly invasive infiltrative growth, no clear aetiology and poor overall survival. A more recent study by Qin et al. [81] showed that PD-L1 expression was higher in HCMV-infected glioma specimens compared to controls, and was mediated by activation of Toll-like receptor (TLR3), a molecule that plays a key role in antiviral recognition and the production of type I interferons. Furthermore, the expression levels of PD-L1 and TLR3 were significantly higher in HCVM-IE positive-gliomas compared with HCMV-IE negative-gliomas [81]. Currently, there are no data associating HCMV infection with increased PD-L1 expression in OC. Nonetheless, the above studies suggest increased PD-1 expression on T-cells and NK cells as well as PD-L1 on glioma cells driven by HCMV infection could negatively impact anti-PD-L1 based therapies such as durvalumab.

## 10. HCMV and Inflammation, a Possible Link for Ovarian Cancer Progression

HCMV infection in monocytes may leads to monocyte activation via activation of NF-κB, the transcription factor that controls cytokine production, and phosphoinositide-3-kinase (PI3K), an enzyme involved in cell differentiation pathways [82,83]. The polarisation and activation of monocytes due to HCMV infection or reactivation leads to high systemic levels of inflammatory cytokines, particularly tumour necrosis factor (TNF) and IL-6, as well as diminished immune function [84,85,86,87,88]. Such an augmented production of proinflammatory cytokines following HCMV infection or reactivation may induce cancer progression directly by promoting undesirable inflammatory conditions in the TME of OC leading to cancer development or suppressed immune function [15,89,90]. Indeed, previous studies have inferred a potential role of inflammatory factors in OC progression as observed in OC patients who have experienced pelvic inflammatory disease [3,91,92,93]. Although clinical HCMV disease is classically manifested only in immunocompromised individuals, there is substantial evidence also that HCMV reactivation occurs frequently in healthy seropositive individuals [88]. Such frequent HCMV reactivation may also exacerbate chronic conditions such as OC.

In OC, approximately one-third of patients present with ascites, a bulging accumulation of fluid in the peritoneal cavity, that is usually associated with resistance to platinum-based chemotherapy, disease recurrence, and poor survival outcomes [94,95]. Studies have shown that OC ascites suppresses the effector functions of both CTLs and NK cells, as it contains abundant immunosuppressive cells, soluble inflammatory mediators and growth factors, all of which helps to promote growth or proliferation of the ovarian tumour [96,97]. A study by Govindaraj et al. [98] showed that CD4^+^ Tregs, particularly those expressing tumour necrosis factor (TNF) receptor 2 (TNFR2), from OC-associated ascites exert more suppressive capacity on effector CTLs than peripheral blood-derived TNFR2+ Tregs. Interestingly, the increased frequencies of TNFR2+ Tregs within the total CD4^+^ T cell pool present in OC ascites was shown to be driven by the presence of pro-inflammatory IL-6 within the ascitic fluid as blockade of IL-6 resulted in reduced frequencies of CD4+TNFR2+ Tregs [97]. Since HCMV reactivation in the TME could lead to increased IL-6 production, these data suggest that active HCMV infection in the TME may also drive the expansion of highly suppressive CD4+TNFR2+ Tregs. Together, these data point towards a potential for an OC immunotherapy targeting IL-6 blockade or anti-HCMV for OC patients presenting with ascites or HCMV in order to improve survival outcomes.

## 11. Conclusions

Presently, there is an overwhelming research interest into personalised medicine, not only in OC but for many other cancers. Both HCMV and OC share similar disease mechanisms through impairment of CTL and NK cell responses thereby promoting their survival or progression. Previous studies have demonstrated that HCMV infects many cell types; promotes tumour growth and induces a pro-inflammatory environment. Furthermore, active HCMV infection creates an immunosuppressive TME that suppress tumour-specific immune responses. To-date, no clinical study has investigated the use of an anti-HCMV approach to treat OC patients infected with HCMV. In GBM patients, however, the use of anti-HCMV targeted T cell therapy has provided some beneficial treatment outcomes, especially in recurrent GBM patients. Therefore, it is reasonable to rationalise that personalised anti-HCMV treatment could potentially help improve the survival outcomes of OC, particularly in patients with active HCMV infection in their TME. Future studies are needed to evaluate the use of anti-HCMV therapy in combination with current established 1^st^ line and 2^nd^ line therapies and evaluate if this combination approach is efficient at improving the survival outcomes of patients with HCMV positive ovarian tumours.

## Figures and Tables

**Figure 1 biomedicines-09-00351-f001:**
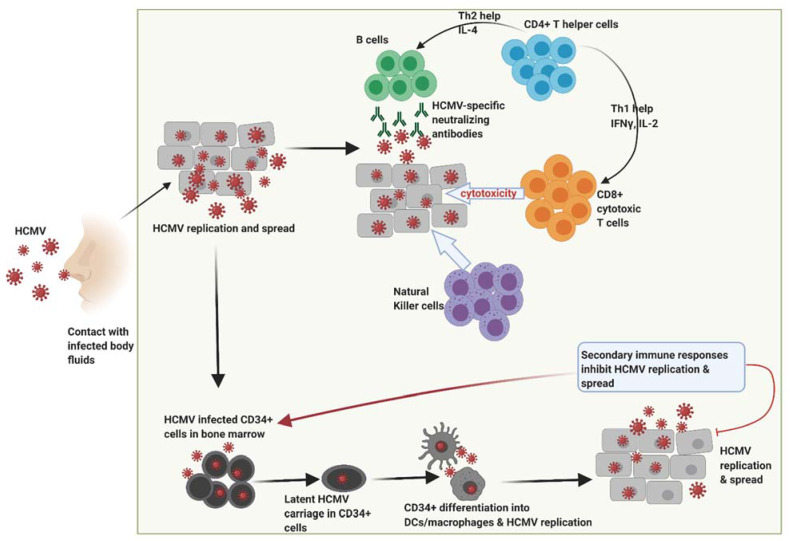
Following primary infection through contact with infected bodily fluids, HCMV replicates within host cells during which time robust immune responses are generated by the host that includes HCMV-specific neutralizing antibodies, natural killer cells and high frequencies of CD4^+^ T helper cells and CD8^+^ cytotoxic T cells. These responses subsequently control viral replication thereby resolving primary infection. However, HCMV has the potential to replicate and spread resulting in infection of CD34+ myeloid cells in the bone marrow and establishment of latency. Differentiation of HCMV-infected CD34+ cells into dendritic cells and macrophages contributes to new HCMV replication during which a secondary immune response induced that helps to inhibit HCMV replication spread. HCMV, human cytomegalovirus. Figure created with BioRender.com.

**Figure 2 biomedicines-09-00351-f002:**
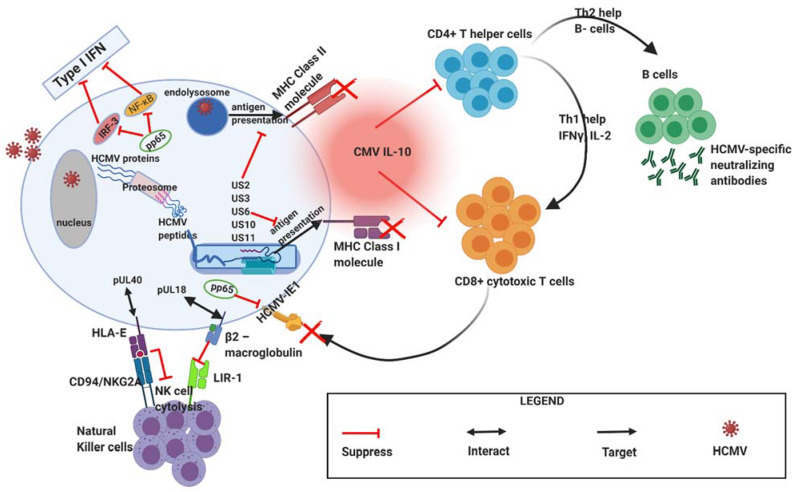
During primary infection, HCMV employs various mechanisms to mediate immune evasion. HCMV expresses viral genes and proteins that interfere with host interferon responses (pp65), inhibit natural killer cell recognition or activation (pUL40 and pUL18), inhibit CD4^+^ and CD8^+^ T-cell recognition by preventing MHC Class I and II antigen processing and presentation (e.g., US2, US3, US6, US10, US11). CMV IL-10 (viral IL-10 homologue) produced by infected cells further acts to suppress CD4^+^ and C8+ T cell responses. HCMV, human cytomegalovirus; MHC, major histocompatibility complex (MHC); US, unique short; UL, unique long. Figure created with BioRender.com.

**Figure 3 biomedicines-09-00351-f003:**
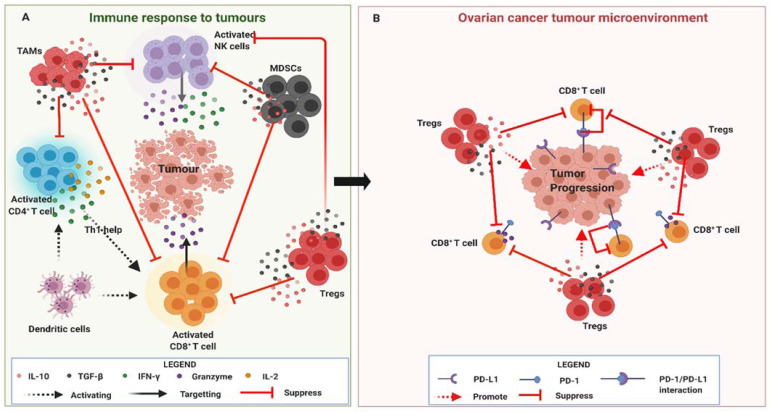
(**A**)The human immune system harbours various immune cells such as natural killer (NK) cells, CD8^+^ T cells, CD4^+^ T helper cells and dendritic cells (DCs) that play an important role in controlling the developing tumour. DCs and CD4^+^ T cells via IL-2 and INF-γ secretion activate CD8^+^ cytotoxic T cells (CTLs) and NK cells which then produce toxic molecules such as granzymes that target the developing tumour. However, pro-tumour cells such as regulatory T cells (Tregs), myeloid-derived suppressor cells (MDSCs) and tumour-associated macrophages (TAMs) produce immunosuppressive cytokines (IL-10 and TGF-β) that inhibit CDTLs and NK cells effector responses. (**B**). The ovarian cancer tumour microenvironment is commonly characterised by high frequencies of infiltrating Tregs and a high Tregs:CD8^+^ T cell ratio that promotes tumour progression. Ovarian tumour also expresses PD-L1 that further inhibits CD8^+^ T cell effector responses upon ligation with PD-1. Figure created with BioRender.com.

**Figure 4 biomedicines-09-00351-f004:**
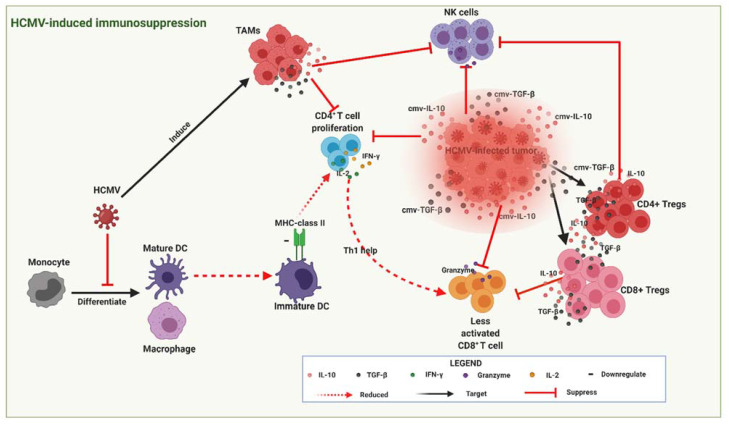
HCMV-infected tumour cells secrete viral cytokines (cIL-10 and c-TGF-β) that create an immunosuppressive environment (highlighted red zone) around the developing tumour thereby promoting tumour progression. cIL-10 and c-TGF-β also induce CD4^+^ and CD8^+^ Tregs that produce similar cytokines (IL-10 and TGF-β) further augmenting the immunosuppressive state by inhibiting cytotoxic CD8^+^ T lymphocyte (CTL) and natural killer (NK) cell effector responses. HCMV also interferes with the differentiation of monocytes into mature dendritic cells (DCs) that leads to the establishment of immature DCs with reduced MHC-class II antigen presentation and subsequently less CD4^+^ T cell proliferation and reduced Th1 help needed for enhanced CTL activation. Figure created with BioRender.com.

**Figure 5 biomedicines-09-00351-f005:**
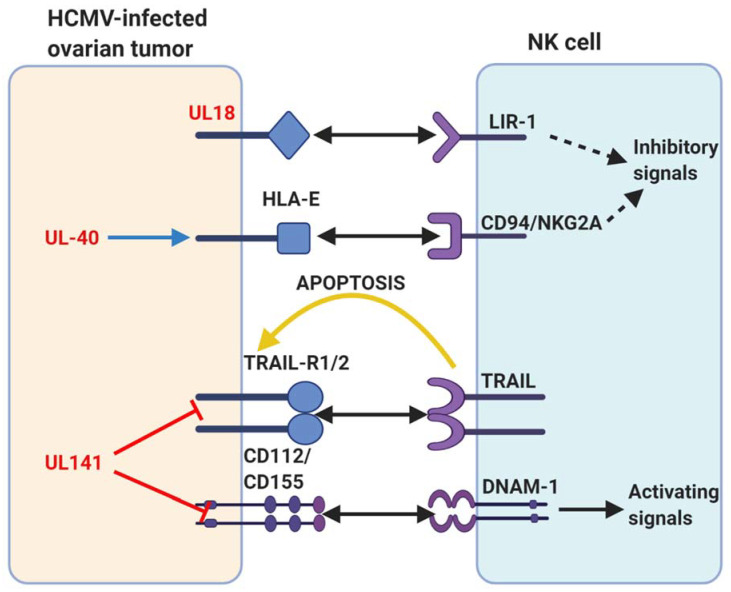
A summary of human cytomegalovirus (HCMV) modulation of natural killer (NK) cell cytotoxicity discussed in this review. Human proteins or receptors are labelled in black; HCMV proteins are labelled in red. The arrows represent actions. Two-pointed solid black arrow = interaction or ligation; one-pointed solid black arrow = intracellular NK cell activation signal; dotted black arrow = intracellular NK cell inhibition signal; yellow arrow = extracellular signal to target cell; red line = inhibits surface expression; blue arrow = increases surface expression. Figure created with BioRender.com.

## Data Availability

Not applicable.

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
