# Peer review of "Potential Impact of Human Cytomegalovirus Infection on Immunity to Ovarian Tumours and Cancer Progression"

_biomedicines, 2021, doi:10.3390/biomedicines9040351_

Round 1
Reviewer 1 Report
In this review, Cox et al., describe the relationship between human cytomegalovirus (HCMV) and ovarian cancer (OC). Although the oncogenic role of HCMV is controversial, there is a growing evidence that this virus may be responsible or at least a contributor to the development of certain cancers including OC. This is a well-written review with excellent figures. However, there are number of formatting mistakes in citing references.
Line 8. Correspondence: is sufficient, remove “Corresponding Author” which is redundant.
Line 58. Study by “Taher, Frisk” should be “Taher et al.,”
Line 140. “Bennet, Glaser” should be “Bennet et al.,“
Line 218,219.”Shanmughapriya, Senthilumar”should be “ Shanmughapriya et al.,”
Line 272. ”Zhang, Ke” should be “Zhang et al.,”
Line 339. “Pesce, Greppi” should be “Pesce et al.,”
Line 348. “Qin, Zhang” should be “Qin et al.,”
Line 380. “Govindaraj, Scalzo-Inguanti” should be “Govindaraj et al.,”
Please include all the authors when you cite the multiple authored articles in the References Section. Do not abbreviate the citation as “ Xie, X et al.,
Please include all the authors in the following references:
3, 4, 6, 7, 9, 13, 14, 15, 16, 17,18,19,20,21,23,24,26,27,28,29,34,35,36,37,38,41,42,43,44,45,49,50,51,53,54,55,56,57,58,62,64,65, 66,67,68,70
Author Response
Please find attached our response.

Reviewer 2 Report
I read with very interest the review entitled “Potential impact of human cytomegalovirus infection on immunity to ovarian tumours and cancer progression”.
This is a well-written, well-organized and well-illustrated manuscript.
Authors discussed about the implication of active infection with HCMV in ovarian cancer progression suggesting that infection-induced chronic inflammation is an essential process for tumours progression. Nowadays, it is important to understand the mechanisms of virus infection control.
Recent advances in the immunobiology of HCMV-host interactions highlight the association between γ marker, human leucocyte antigens and killer immunoglobulin-like receptors and the natural course of human cytomegalovirus infection. So, in order to improve the quality of manuscript, I suggest to discuss about this recent evidence and speculate a possible implication in ovarian cancer. Please complete the list of references with the following excellent articles: PMID: , PMID: 29067686; PMID: 30764515.
Therefore, I believe that the present paper should be accepted after minor revision.
Author Response
Please find attached our response.
